# Agroforestry Innovation through Planned Farmer Behavior: Trimming in Pine–Coffee Systems

**Edi Dwi Cahyono [1],\*, Salsabila Fairuzzana [1], Deltanti Willianto [1], Eka Pradesti [1], Niall P. McNamara [2], Rebecca L. Rowe [2] and Meine van Noordwijk [3]**

[1]  Faculty of Agriculture, Universitas Brawijaya, Malang 65145, Indonesia; salsabilazatira@gmail.com (S.F.); deltantiwww@gmail.com (D.W.); eka_pradesti@student.ub.ac.id (E.P.)

[2]  UK Centre of Ecology & Hydrology, Lancaster LA14AP, UK; nmcn@ceh.ac.uk (N.P.M.); rebrow@ceh.ac.uk (R.L.R.)

[3]  World Agroforestry (ICRAF), Bogor 16155, Indonesia; m.vannoordwijk@cgiar.org

\*  Correspondence: edidwi.fp@ub.ac.id

**Abstract:** Knowledge transfer depends on the motivations of the target users. A case study of the intention of Indonesian coffee farmers to use a tree canopy trimming technique in pine–based agroforestry highlights path-dependency and complexity of social-ecological relationships. Farmers have contracts permitting coffee cultivation under pine trees owned by the state forestry company but have no right to fell trees. A multidisciplinary international team of scientists supported farmers at the University of Brawijaya Forest in East Java to trial canopy trimming to improve light for coffee production while maintaining tree density. Data were collected using surveys through interviews, case study analysis using in-depth interviews, focus group discussions and nonparticipant observations. Using the Theory of Planned Behavior, we found that though farmer attitudes toward trimming techniques were positive, several factors needed to be scrutinized: perceived limited socio-policy support and resources. While there is hope that canopy trimming can improve coffee production and local ecosystem services, a participatory and integrative extension and communication strategy will be needed. In the relationship between farmers as agents and forest authorities as principals, any agroforestry innovation needs to incorporate knowledge and concerns in the triangle of farmers, policymakers and empirical science.

**Keywords:** agroforestry; innovation transfer; trimming; intention; participatory and integrative research-extension; stakeholders

## 1. Introduction

Innovators—and especially developers of innovations that others are supposed to use—may assume that once innovations are transferred they are inherently beneficial and accessible to the potential adopters [1–4]. In other words, they are subject to a "pro-innovation bias". By starting from the needs of the targeted adopters a different design of an innovation process may lead to alternate outcomes [5–8]. How benefits and costs of any innovation are distributed is key to the acceptance of innovations, but groups, rituals, affiliation, status and power are major drivers of human decision-making in a social context [9]. In a context of agroforestry, trees and farmers interact, at the plot, landscape and policy scales [10]. Ecological interactions between trees and other agroforestry components, both complementary and antagonistic, can be a mirror for the social interactions between forest institutions and farmers. This is especially true in a specific form of agroforestry where forestry institutions own the land and are interested in permanent tree cover and timber harvests. While farmers may benefit indirectly or directly from timber production, they are primarily benefiting from and

interested in understory crops leading to potentially antagonistic relationships. The principal–agent theory, as developed in institutional economics, analyses the strategic interactions between principals who delegate, often limited in time or scope, a conditional grant of authority to an agent [11,12]. An agent (e.g., farmers or farmer groups) is an individual or entity that has the authority to act on behalf of the principal (party or parties, e.g., forestry institution) who has power over the agent. A dispute—namely the principal–agent problem—arises when the actions of agents pursue different interests or even are in conflict with those of the principal [12]. Forms of "social forestry" (locally known as "tumpangsari" and internationally as "taungya") as implemented on densely populated Java (Indonesia) have such "principal–agent" interactions between the forestry principals, who under the social forestry program have granted increased authority to farmer to act as "agents" for the "principal" [13]. This principle–agent interaction faces the inherent challenges of this type of relationship, with little harmony in management objectives expected and strict enforcement of forestry rules essential, but not always feasible leading to the risk of agent shirking and slippage. Analysis [14] suggests that other types of benefit-sharing arrangements are needed before a much wider range of locally adapted solutions can be found, which more fully exploit the opportunities of tree–crop synergy. Social forestry programs in Indonesia have tried to bring more balance to the stakes of the forestry principals and farmer agents, including timber sharing arrangements [15]. However, technical innovation may still be constrained by the institutional context and its history (path dependency), as our case study explores.

For many decades, the interaction between Indonesian forest authorities and forest-edge farmers who have tried to expand their coffee into existing forests has been strongly antagonistic in Indonesia, as also evident elsewhere [16–18]. Only after the 1997 "reformasi" period and the 1998 forestry law, the "tumpangsari"—contracts that allow farmers to grow annual food crops among newly planted trees for a few years—were extended to allow longer-term shade-tolerant perennials such as coffee or fodder for a cut-and-carry system to be grown [19–21]. For the state forest company authorized to manage these lands, the survival and growth of the tree plantation has been the priority and farmers risked being left out of future contracts if trees were found to have been damaged. The intercrops were contractually treated as a "share-crop", with around one-third of the yield belonging to the "landlord," (in this case, the state forest company). These rules gave the state forest company a stake in the productivity of intercrops (different from when a fixed "land rent" would have been charged) but reduced the profitability for the farmer. Any intervention, such as social innovations from forest authorities or technical innovations from development agencies can destabilize the agent–farmer economic structure and cause disputes with the existing authority-principal policies, which cause resistance towards agroforestry innovation [22,23]. A balance needs to be found between agents (or principals) who may be profiteers, prove-it accountants or visionary prophets [24,25]. Profiteers are profit agents (entrepreneurs) as part of the national economy; prove-it accountants are intermediary agents that encourage transparent processes for mutual benefit. Prophets see themselves as representative of the local community to voice their interests, needs and perspectives in the complexity of the external environment. The roles of NGOs (prove-it accountants) as an intermediary for marginal groups to negotiate with the authorities (profiteers) have been identified [26,27]. On the other hand, efforts to put forward local forest community interest ("the prophet") form a challenge [28] that open spaces for more a deeper study.

Our case study focuses on an area of forest on the slopes of the Arjuna volcano in Malang regency, Java where principal–agent interactions had been ongoing for two decades. The forest management was handed over to Brawijaya University (UB) to be operated as an "education forest" and the new university management team proposed a change in forest-farmer relations. Rather than trying to collect the coffee share crop revenue directly, they asked all farmers to sell their coffee (purportedly for going market prices) to the new university UB-Forest Management company. It was anticipated that through better coffee processing and targeting other market segments enough profit would be made to offset the foregone sharecropping payments. UB-Forest management was also looking to return

profit or benefit to the farmers in other ways, e.g., community electricity provision which may not be fully recognized or thought valuable by the farmers. In practice, farmers quickly reverted to selling most of their coffee to outside traders, as they could achieve a better price than selling to the UB-Forest Management company.

As part of the wider interaction between UB and the forest community, UB researchers independent of the UB-Forest management team proposed to address technical constraints around farmer yield and income (Figure 1). One such constraint was the low levels of light for the promotion of healthy understory coffee growth due to the overlying dense pine forest canopy. Our overarching hypothesis was that while farmers perceived the introduced innovation positively, some individual and external factors may hamper their intention to practice it.

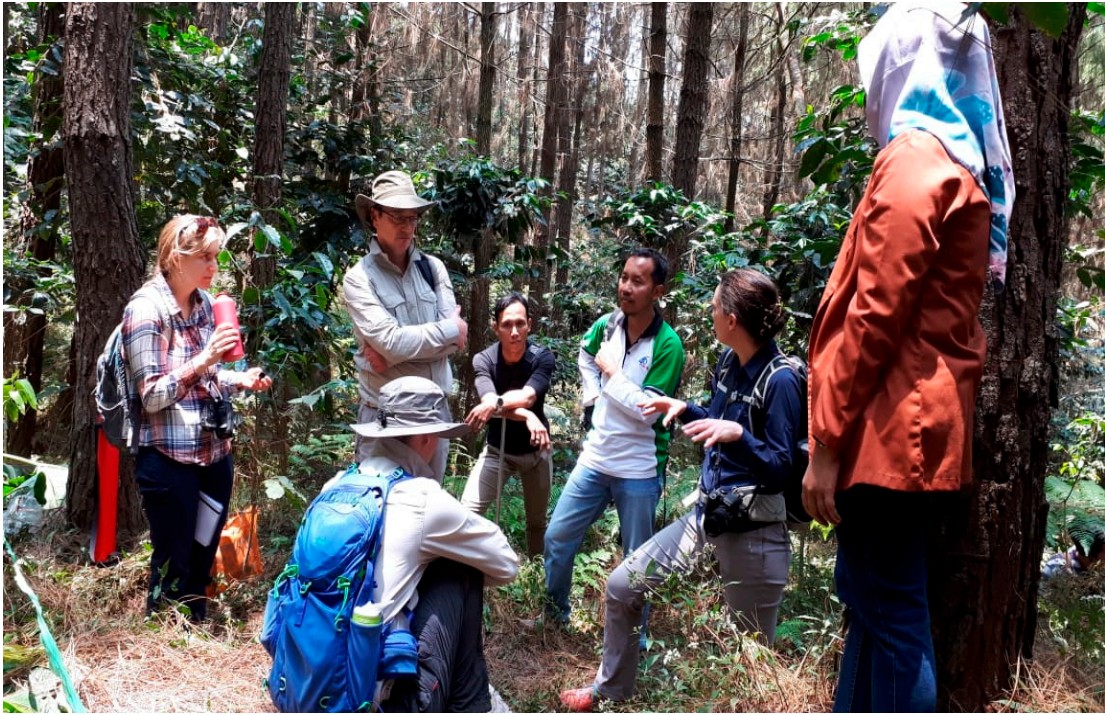

**Figure 1.** Coffee plants under the pine forest canopy and the interdisciplinary science experts.

Our specific research questions were (1) what farmers thought about innovative efforts to provide more light to understory crops, (2) how this relates to their knowledge and expectations and (3) under what conditions it could be sufficiently profitable for them to undertake this innovation. We explored these questions with reference the theory of planned behavior, as we will describe here, before providing details of the site and specific research methods.

## 2. Methods

### 2.1. Theory of Planned Behavior

The theory of planned behavior (TPB) (Figure 2) can reveal the cognitive, affective and motivational dynamics of farmers facing new techniques. TPB is commonly employed in the fields of health and economics [29,30], but rarely in the field of agroforestry, agriculture or environmental management. A recent study using TPB in such a field was carried out in Serbia [31]. Some researchers argue that self-efficacy is interrelated with the three antecedent variables to measure intentions [32,33]. For this reason, we included self-efficacy as part of the variable of perceived behavioral control.

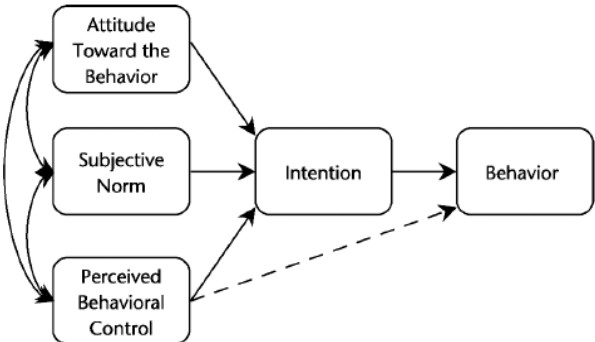

**Figure 2.** Theory of planned behavior, adapted from [32], showing the three antecedent interacting aspects: (1) attitude towards behavior/behavioral beliefs; (2) subjective norms/normative beliefs; and (3) perceived behavioral control, intention and behavior/action.

The single construct of intention may be compared with the three antecedent variables of intention. In addition, as can be seen in the model, Perceived Behavioral Control (PBC) is a unique variable that acts as an antecedent for intention and actual behavior as well. Intention is interpreted as volitional control or willingness to take certain actions [34–36]. In this study, intention variables were treated as the consequence or impact of attitudes, subjective norms and perceived behavioral control. We also measure intention solely (as a single variable), to be exemplified farmer intentions in applying trimming techniques; the score can be compared with the three antecedent variables of intention.

Typically, previous studies used "perception" and/or "attitude "as a determinant of adoption [37–40]. The TBP allows the variables of attitudes, subjective norms, and, perceived behavioral control to be combined as determinants of intention [41,42]. Furthermore, the self-efficacy variable (self-confidence) may be added to TBP to predict intentions. Hence, TBP can describe the behavior that is influenced by external factors (nonvolitional behavior) and self-efficacy as the individual's internal factor [43].

In the context of agroforestry, we expect that an intention to use pine trimming is associated with a positive attitude (expectations of benefit), strong social support and perceived low constraints of farmers to implement such techniques. In practice, the possible low scores of intention variables can lead to the rearrangement of social and communicative interventions to increase the adoption of trimming techniques. Given the current challenges and opportunities in the UB-Forest, there are simultaneous efforts to explore: (1) techno-ecological options for pine–coffee agroforestry revitalization (such as increasing light intensity, improving crop and soil management (trimming leaves used as litter/organic fertilizer); (2) socioeconomic options (the integration of stakeholders, such as to the financing of production factors); and (3) renewing the existing Indonesia's forestry policy framework to overcome past conflicts (sharing systems, alternative marketing channels and land management regulations) and achieving effective joint management (involving farmers, local governments and forest authority).

## 2.2. Place, Location and Research Participants

Currently, a variety of collaborative schemes on community forest management exist in Java. The collaboration is between forest management and members of the surrounding communities. UB-Forest management inherited contracts made by the state forest company in response to massive forest land annexation by the community at the end of the 20th century due to political reforms. Cultivation of coffee under pine (or mahogany) trees is common in UB-Forest. At the end of 2015, The UB-Forest management through the University of Brawijaya (UB) received the mandate to administer the forest on behalf of the Indonesian Minister of Forestry and Environment. This production forest—which is located on the slopes of Mount Arjuno (altitude 1200–1900 m in the Karangploso District, Malang regency) has three specific objectives: education, research and ecotourism. Historically,

the pioneer farmers settled around the UB Forest in the 1920s and the present farmers are the descendants from the early farmers. Their livelihoods are highly dependent on the agroforestry system, in which they also cultivate food crops and vegetables along with the coffee under the pine trees.

The research location was the Sumbersari Hamlet in the enclave area of UB-Forest (Figure 3).

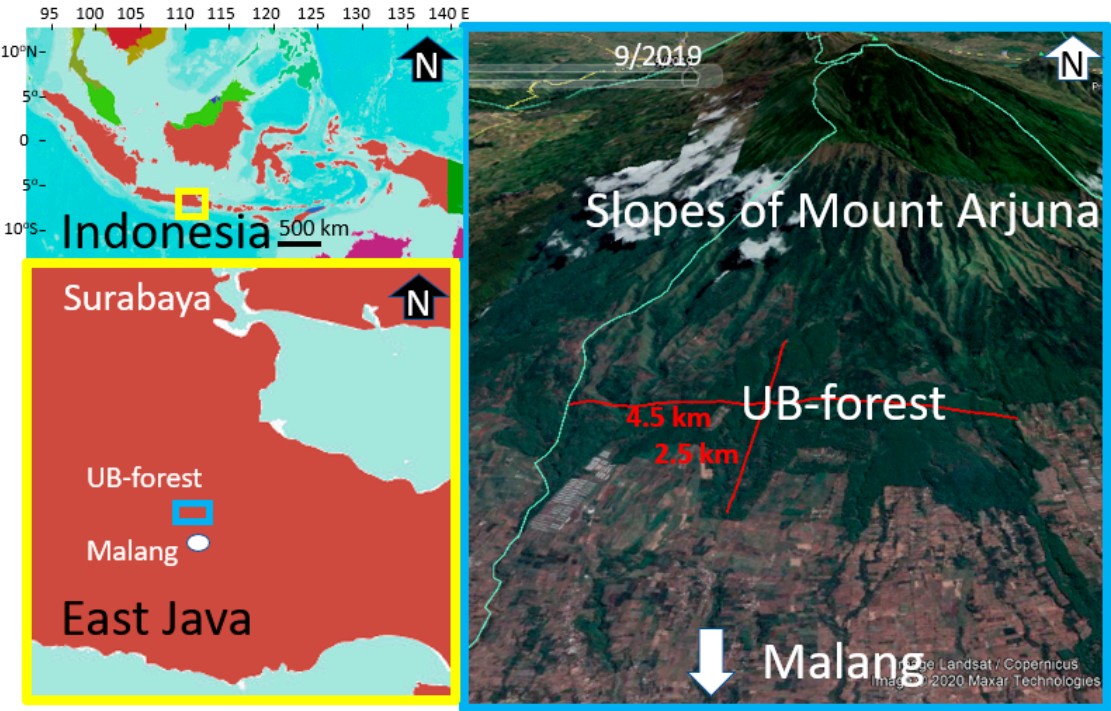

**Figure 3.** Research location UB Forest in East Java, Indonesia.

The hamlet is located in the Kalisari Watershed of Malang Regency, East Java. Data were collected in the period of March–May 2019. The senior researchers were those from the fields of agricultural extension/communication, soil science and agroforestry. Communication was established with a local key person to get wider access to the sites and respondents. Approval for this study was obtained from the management of UB-Forest and local regional authorities. Anonymity and voluntary participation of the participants were ensured. The research participants for the survey comprised 22 farmers out of an overall 30 within the hamlet, who were cultivating coffee under the pine. The research participants were verbally informed about the purpose, use and possible impacts of the research; in the local context written consent was not deemed to be required. Several members of the hamlet, including its leader, were involved in testing the innovation in the forest.

*2.3. Research Design and Methods*

The research design used the four stages of the action research approach (Figure 4): plan, act, observe and reflect [44]. It used mixed methods, during the observation stage combining quantitative (through a survey) and qualitative data collection (case study).

Plan—Preparations, site visit. Researchers from UB-UKCEH-ICRAF conducted site visits and a series of discussions under the umbrella of the agroforestry research group at UB to identify issues to be solved through action research. UB and UKCEH accessed funding through different research funding schemes to support the project.

Act—Intervention on-site and communicate with farmers. To overcome low coffee productivity under the pine trees, canopy pine trimming techniques were disseminated. This technique was considered new, as it had never been applied by local farmers. Strategies to communicate this innovation was by (1) "advocacy" through personal communication to a key farmer as the gatekeeper

to gain trust; (2) "behavior development" through group meetings with farmers followed by a demonstration of pine trimming techniques on the site. The communication for the advocacy strategy was designed to ensure the key farmer understood the intended benefits of using the technique. Then, he acted as the "innovation endorser"—influencing other farmers to adopt the technique. For this reason, he was also asked to demonstrate the techniques to his peers.

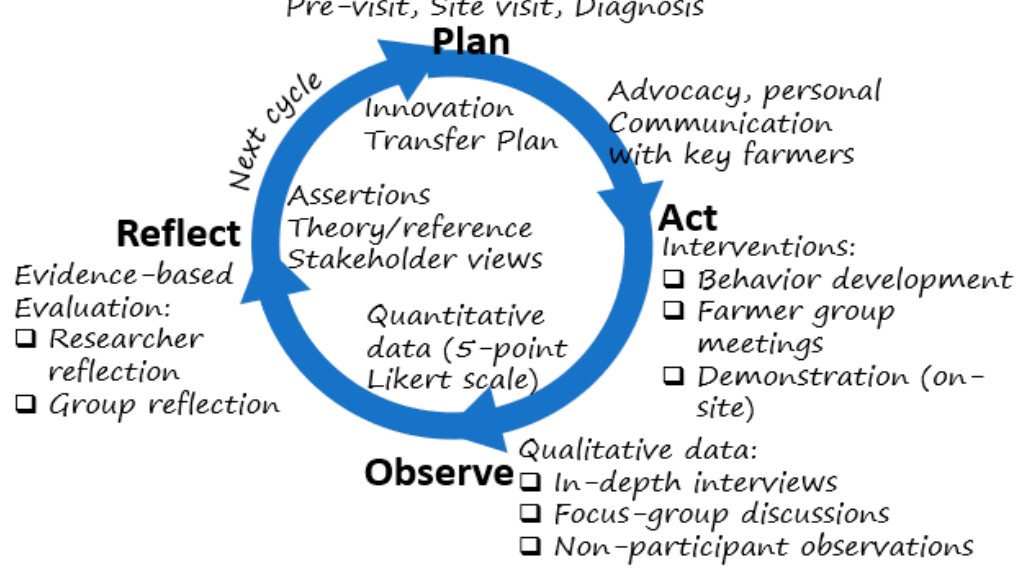

**Figure 4.** Stages of action research for transferring the trimming techniques.

Observe—Collection data on sites/evaluate evidence. Farmer responses were measured using a mixed-methods sequential explanatory design [45]. This design covers two phases: (1) quantitative data collection through a "survey" of the coffee growers (executed March—April 2019); (2) qualitative data collection using a single instrument case study (May 2019) regarding the grower perceptions of the trimming technique in the context of UB Forest management. The qualitative data collection helped to get a deeper understanding of the quantitative database. This was especially important when dealing with extreme cases or demographic factors that may be associated with farmer intentional behaviors.

Reflect—Reflection was based on the research findings/evidence, which was applied through (1) individual/researcher evaluation; (2) discussions between researchers; and (3) stakeholder views through research finding seminar (involving resource person; a representative from university, governmental agents, NGOs, etc.); this is also a way to increase external validity. Subsequently, assertions/affirmation was conducted by clarifying the findings using the theory of planned behavior and other references and followed by giving options for further study and intervention approach.

### 2.4. Research Instruments

For the survey, a closed questionnaire (quantitative database) was administered to all participants (22 farmers), covering variables of intention that were (1) attitude toward trimming (6 indicators, e.g., covering statements such as "I believe that pine trimming . . . is necessary because the litters can become mulch"; (2) subjective norms (8 indicators, e.g., "In my opinion . . . local community supports trimming"; and (3) perceived behavioral control (9 indicators, e.g., "I am confident to trim, because . . . it is easy to trim at 10 m high". The five-point Likert scale technique was employed: "strongly disagree" (score 1), "disagree" (2), "neutral" (3), "agree" (4) and "strongly agree" (5). The statements were based on theoretical concepts and adjusted with pine–coffee agroforestry issues as exposed by a key farmer (Appendix A); Additional questions were about farmers" demographic status (age, education, farming systems and experience, source of coffee farming information). The related questions for in-depth

interviews can be seen in Appendix B. To reduce nonresponse error, respondents were interviewed face-to-face to overcome their unfamiliarity to fill out the questionnaire.

*2.5. Data Validity*

For the quantitative data, content validity and reliability tests were managed by experts, discussions with farmers and statistical tests. The steps to develop indicators for intentional aspects: (1) the variables were derived from published theories [46–48] and indicators arranged based on discussions with peers and leading researchers and (2) from extracting key concepts from key farmers during initial interview sessions. The validity of the statement items in the questionnaire was justified by members of the research team. Questions were then tested for clarity in a pilot study with 10 farmers following which sentences within the questionnaire were revised to increase participants" understanding. Statistically, item validity is measured by the correlation coefficient between an item's score and the total score of all items at a certain significant level (10%). The validity of the intention related items was tested with the SPSS statistical package resulting in a Pearson's correlation value (r) of > 0.36, indicating the items have high validity.

As a measure of the reliability of items, internal consistency was also tested. This technique examines the correlation between multiple items to measure a construct. The calculation of Cronbach's alpha Coefficient was rendered for item sections in the TBP questionnaire. The high instrument reliability was indicated by a coefficient of more than 0.80; any item coefficient of less than 0.30 need to be eliminated to avoid an unreliable indicator [49]. The Cronbach's alpha value was 0.85, indicating high reliability of the items.

*2.6. Data Collection Methods*

As a follow-up, after analyzing quantitative data, the researchers interviewed key farmers again to cross-check noteworthy quantitative data (such as the mode or extreme data) as a case for presentation (embedded analysis) [50]. Other techniques were used to triangulate: (1) nonparticipant observation (by observing the agroforestry setting/location); (2) member-checks" that is discussions with the key farmer and other farmers regarding the findings; (3) focus group discussions (FGD). The first FGD was conducted with 12 coffee farmers at the local meeting house to discuss the technical and social aspects of pine–based agroforestry cultivation. The second FGD was carried out in the local village meeting hall (*Balai desa*), by inviting key farmers (those who were the first to grow coffee or "the innovators") and coffee farmer representatives (those who planted coffee afterward or "the followers") and the stakeholders (representatives of the neighborhood and citizen associations, village officer, rural forest community member, village and sub-district officers, UB Forest employee, Perhutani/state-own forestry company agent and UB agroforestry research group members). In this activity, farmers were welcome to express their views on various issues, such as the relationship between the local community and the forest management authority, the history and quality of partnership, farmer economy and policy. These steps were useful to explore and describe sensitive issues and complexity of farmer livelihoods as the context which may influence their intentional behaviors. Field notes, camera imagery, and audio recordings were used to record the data. The data were recorded on a CD for storage only while useful for further intended research. Data are presented in an anonymized way to protect respondents.

*2.7. Data Analysis*

For the data analysis, descriptive statistics (frequencies, percentages, mean scores and standard deviations) were calculated to summarize the survey data. Separate tabulation was applied to capture qualitative data. The qualitative findings (statements, observation, photos) being used to validate and provide context to the qualitative results. Research findings were also presented in a series of seminars with members of the research team and stakeholders as part of the reflection and interpretation process.

## 3. Results

### 3.1. Social Context and the Demographic Factors

The agroforestry land in the UB-Forest area managed by the respondents of this study totaled 33.04 ha and the total area of UB-Forest is 544 ha. All respondent-farmers managed crops both in UB Forest 13.5 ha (pine-coffee agroforestry), but also in the state-owned forest (Perhutani) 18 ha (pine-vegetable agroforestry) and other forested areas 1.5 ha (pine-vegetables). Although, the survey data shows that the average of land areas managed by the respondent was relatively limited, at 1.2 ha (in Perhutani land); 0.7 ha (in UB Forest) and 0.5 ha (in farmer home gardens). The demographic data (Figure 5) show that 15 of 22 farmers (68%) have been farming within the agroforestry scheme for more than 20 years. Most respondents (41%) were in the >50-year age class and 77% of the farmers have had limited access to education being educated only to the elementary school level.

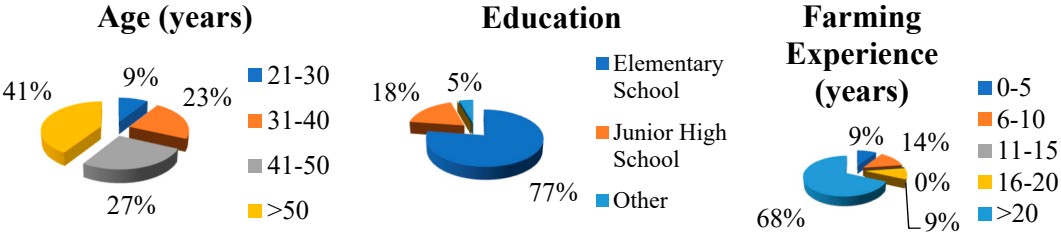

**Figure 5.** Demographic characteristics of respondents (n = 22).

Moreover, living in relatively remote areas (enclave) had become a constraint for their economic development. The results of an interview with a farmer revealed that in the past (especially before the reform era) the forest authorities did not allow farmers to grow coffee under the pine canopies, arguing that "forests cannot be turned into gardens." The coffee planted by farmers as an agroforestry system was the result of farmers" struggles to survive in these enclaved forest areas, despite causing conflicts with the forest authority at that time. In a lower level of preference, farmers also cultivate other types of crops, such as horticulture and food staples in sufficient quantities for their daily needs. The current use of chemical fertilizers from outside is still difficult as before, though they need it. One farmer stated, "apart from being far away from obtaining fertilizer, it is also expensive," indicating their high dependence on local efforts to obtain adequate coffee yields.

We asked the respondents about their experience in accessing the information on coffee cultivation. The data on Table 1 shows that 19 of 22 respondents (86%) had participated in extension activities discussing coffee cultivation. The sources of information were from public agricultural workers and/or from Perhutani personnel or experts from UB. Only 4 of 22 farmers (18%) received information on coffee cultivation from farmer groups and 7 of 22 (32%) received the information from other sources.

**Table 1.** Transfer of information on coffee cultivation (n = 22).

| No | Source of Information | Number of Respondents Receiving the Information |
|---|---|---|
| 1 | Office of agriculture | 19 |
| 2 | State forest Company/Perhutani | 19 |
| 3 | University staff | 19 |
| 4 | Forum/farmer Group | 4 |
| 5 | Others | 7 |

### 3.2. Attitudes toward the Trimming

As has been noted, the primary aim of this study was to measure the intention levels of farmers to adopt pine trimming techniques. We calculated the supporting variables: attitudes toward the trimming, subjective norms (social supports) and perceived behavioral control (obstacles in doing

trimming) using a Likert scale from one (strongly disagree) to five (strongly agree). The study showed that 79% of respondents agree or strongly agree that pine trimming "provides benefits" for farmers (item #5; Table 2). The perceived benefits primarily related to the pine litters being transformed into mulch (item #1); and to a lesser extent benefits were expected to arise from fertilizing the soil, increasing light at the level of the coffee canopy or use of branches as firewood (items #1, #3 and #6). Moreover 83% of respondents agreed or strongly agreed that trimming can "increase coffee production" (#2).

**Table 2.** Attitudes towards pine trimming (n = 22), value based on a Likert scale from one (strongly disagree) to five (strongly agree).

| No | Statements: I Believe that Pine Trimming … | % "Agree" or "Strongly Agree" | Means (M) of Attitudes | Standard Deviation (Sd) |
|----|----|----|----|----|
| 1 | ■ … is necessary because the leaf litters can be made into mulch. | 84.55 | 4.23 | 0.53 |
| 2 | ■ … increases coffee production. | 82.73 | 4.14 | 0.64 |
| 3 | ■ … fertilizes the soil. | 81.82 | 4.09 | 0.68 |
| 4 | ■ … increases the incoming light. | 81.82 | 4.09 | 0.87 |
| 5 | ■ … provides benefits. | 79.09 | 3.95 | 0.72 |
| 6 | ■ … is useful because the branches can be used as firewood | 78.18 | 3.91 | 0.68 |

It is worth noting that the mean score of "provides benefits" (#5) is slightly lower than most of the other indicators (except benefits of getting firewood), indicating the possibility that the farmers are not just considering the direct "benefits" to coffee yields, as is the objective of the question, but are instead considering the balance between benefits and risks (Table 2).

As seen in Table 2, there is a considerable variation in the perception scores. An in-depth interview with a farmer revealed some uncertainty as to the level of benefit of pine trimming, "If it is only trimmed, then it does not have much effect … only the number of leaves on the coffee plants increases." This indicates that although considered generally useful, there is an uncertain view of the trimming technique.

*3.3. Subjective Norms of Trimming*

Farmers were asked about the level of social support related to pine trimming, which reflects the subjective norm variable (Table 3). When confronted with the statement "experts help farmers know how to trim through demonstration" (item #2), 81% of respondents said agree or strongly agree. This is consistent with the response to the statement "I understand the discussion about trimming (#1): 83% expressed agree or strongly agree. To a lesser extent, there were only 75% of respondents who agreed or strongly agreed that "hamlet personnel support trimming" (indicator #8). It is interesting to note that a relatively moderate response was given regarding the perceived support by UB-Forest management, both in terms of support for trimming and especially for land maintenance (indicators #5 and #7).

The findings show that respondents felt that they received support not only from UB-UKCEH-ICRAF personnel, but also from the local community and their families. On the other hand, the support from the UB-Forest management and local officers (hamlet personnel) were moderate or even considered limited. This is likely because of perceived problems of the newly revised coffee sharing mechanism and also possible latent conflicts in the community. Moreover, a complicated relationship was experienced by farmers with *Perhutani*, the former state forest management entity.

**Table 3.** Subjective norms/social supports of pine trimming (n = 22).

| No | Statements: In My Opinion … | %"Agree" and "Strongly Agree" | Means/M of Subjective Norms | Standard Deviation/Sd |
|---|---|---|---|---|
| 1 | ■ … I understand the discussion about trimming. | 82.73 | 4.14 | 0.35 |
| 2 | ■ … experts help farmers know how to trim through demonstration. | 81.82 | 4.09 | 0.53 |
| 3 | ■ … the local community supports trimming. | 80.91 | 4.05 | 0.38 |
| 4 | ■ … my family supports trimming. | 80.91 | 4.05 | 0.58 |
| 5 | ■ … UB Forest supports trimming. | 79.09 | 3.95 | 0.58 |
| 6 | ■ … discussions with friends encourage me to look for additional information about trimming. | 78.18 | 3.91 | 0.81 |
| 7 | ■ … UB Forest takes care farmers to take care of the agroforestry lands. | 76.36 | 3.82 | 0.40 |
| 8 | ■ … hamlet personnel support trimming | 74.55 | 3.73 | 0.70 |

A key farmer expressed the historical context plainly:

*"These coffee plants … in [the year of] 96 was very difficult because at that time the Perhutani did not allow us to plant coffee under the shade … under the stands of pine."*

*"The reason was that the forest cannot be used as a garden. However, I think because we live in a forest area … the impression is that it destroys the forest, but we must earn a livelihood for our survival."*

*"For this reason, I tried to plant whatever crops I planted at that time. The plant that was able to grow and produce just coffee. It means that the people here are still looking for income in the forest area."*

Hence, this study shows that farmers receive various levels of support from different parties. Relatively high support from the UB-UKCEH-ICRAF is probably because of the action of disseminating the trimming techniques through site demonstrations and discussions with farmers on some occasions.

*3.4. Perceived Behavioral Control of Trimming*

We noted that perceived behavioral control influences farmer intentions to undertake to trim. We use two primary variables to measure this self-efficacy (individual factors) and non-self-efficacy (other factors). The indicators of self-efficacy covering aspects of farmer confidence to trim because of expected improvements in their livelihoods, easiness of trimming techniques or suitability with local conditions, etc. Meanwhile, the indicators of non-self-efficacy include the availability of finance, time, information or other resources. The results are presented in Table 4.

When confronted with the statement "if funds are available" (item #1) and "the demonstration by experts encourages me to trim" (#2), 83% and 80% of respondents stated "agree"/A and "strongly agree"/SA, respectively. Furthermore, the number of respondents who strongly agreed or agreed however is reduced when responding to the statement "it helps improve my life" (#3); and "it is easy to understand" (#4).

**Table 4.** Perceived behavioral control/PBC of trimming (n = 22).

| No | Statements "I Am Confident to Trim … " | %"Agree" and "Strongly Agree" | Means/M of PBC | Standard Deviation/Sd |
|----|-----|-----|-----|-----|
| 1 | ■ … if funds are available. | 82.73 | 4.14 | 0.35 |
| 2 | ■ … because the demonstration by experts encourage me to trim. | 80.00 | 4.00 | 0.76 |
| 3 | ■ … because it helps improve my life. | 77.27 | 3.86 | 0.64 |
| 4 | ■ … because it is easy to understand. | 76.36 | 3.82 | 0.85 |
| 5 | ■ … because I will do trimming in the farming land. | 76.36 | 3.82 | 0.91 |
| 6 | ■ … because trimming is suitable with local condition. | 75.45 | 3.77 | 0.69 |
| 7 | ■ … because it is easy to do. | 70.00 | 3.50 | 1.10 |
| 8 | ■ … because it is cheap. | 67.27 | 3.36 | 1.05 |
| 9 | ■ … it is easy to trim at 10 m high. | 56.36 | 2.82 | 1.05 |

The more statements cover aspects of follow-up or practicality, the less the farmer's intention to do the trimming ("I will do trimming in the farming land": agree or strongly agree = 76). Less than 77% of respondents agree or strongly agree to practice the trimming. To exemplify suitability to local conditions (#6); availability of funds (#8) and time (#7). These responses are also consistent with a finding in our study, in which 75.45% of respondents stated "my time is limited to trim. "The lowest agreement is for the ease of trimming at a height of 10 m (#9), indicating the potential safety risk of this work (Figure 6). The key farmer highlighted this case:

> "*As you get older … you only dare [to go up] in part … If you go up maybe up to three to five trees you can still be brave, but not for the whole day … If more than that [five trees] I start to tremble … often slides. It means that the conditions are not possible to carry out all day … even though one hectare has more than 600 trees.*"

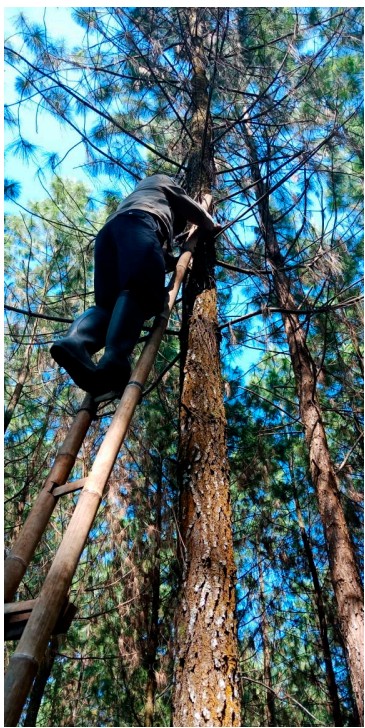

**Figure 6.** Demonstration of pine trimming by a key farmer.

Only a moderate number of farmers (76%) "will do trimming in the farming land" (#5, M = 3.82; Sd = 0.91). However, further analysis shows that proper extension methods may increase the farmer's intention to seek additional information—or apply trimming. Among these methods were interpersonal communication, such as discussions with friends on pine trimming (#6); especially discussions with the experts; and provision of demonstration plots (#2).

Last, but not least, there are strong indications that external factors were also key points to raise the level of farmer intention. The statement "if funds are available", then they will be confident to trim (A/SA = 83%). Similarly, when the trimming is associated with other external resources, such as local condition (#6), easiness to do (#7), money (#8) and potential risk (#9) (agree or strongly agree < 76%). These indicators suggest the conditions that may hamper the application of trimming, as expressed by a farmer: "To be honest, if I trim it, I'm not happy ... the pine remains small, but high. Aside from work safety, the risk is falling ... "*suloyo*" [local term for "sluggish, weak or helpless"]. Don't let it happen."

## 4. Discussion

The process to change individual behavior, commonly seen in an innovation/technological transfer, tends to pro-innovation bias [3]. The assumption is that new technology is better than the previous one. One of the problems is that for targeted individuals the nature of technology is perceived as complicated, as is evident in this study. Nevertheless, our findings have compiled evidence that shows trimming is complex from a farmer's perspective. An individual's intentions are also influenced by considerations of individual capacity, including technical complexity and resource availability to operate. This is particularly evident when such innovation is interfaced with marginal farmers, as in the case of pine–based agroforestry community in Java. We have used the theory of planned behavior as the lens to understand farmer intention to undertake the innovation of pine canopy trimming. We found that the value of the three factors of intention varied.

The attitude/behavioral beliefs—Measurements of the theme or aspect of attitude indicate that, to some extent, the economic benefit is an important element in forming farmers" attitudes towards pine trimming. However, income stability and production costs are also significant factors, regardless of plant productivity. The additional cost for trimming could be a sensitive factor as limitations of family labor and the age of farmers would necessitate the employment of outside labor to complete the trimming. The sensitivity of the extra costs for a small enterprise has long been identified as a sensitive factor for marginal farmers [50].

From the ecological perspective, trimming is perceived useful for (1) increasing light intensity for coffee plants; and (2) fertilizing and mulching the soil to retaining moisture. The literature shows the significant roles of mulch for improving microclimates for plant growth while suppressing weeds [51,52]. Retaining soil moisture is particularly valuable for dry land agroforestry such as in the UB Forest. Therefore, from the techno-ecological views, farmers are responsive to the innovation; their motivation is to increase productivity under the pine trees—vegetables, food crops, and especially coffee plants. The current pine canopy has suppressed coffee growth; the optimum light intensity should be 60–80% [53] but is lower in UB due to over-shading by the pine. This can result in the mortality of coffee tree branches and problems of shortening the harvesting period [54,55].

The subjective norms—We have noted that subjective norms are a belief to fulfill the social expectation. The support from the UB-UKCEH-ICRAF is shown in the trimming demonstration site to help farmers' awareness of and interest in the trimming. However, limited assistance from the UB-Forest management, possibly because of limited involvement at the early phase of the project, is a potential challenge although engagement with this UB-UKCEH-ICRAF project is now ongoing. Moreover, farmers are unhappy as their previous proposals to the forest management to get agricultural inputs were neglected, as expressed during an interview session. Farmers have never practiced trimming, perhaps because of the perceived strict rules on the treatment of the pine tree by the former

and current forest management. Hence, the trimming technique itself may be simple in practice, but there are socio-psychological barriers due to the considerations of past rules.

Perceived behavioral control—As mentioned, self-efficacy is a belief to do something. We use this concept to measure farmer confidence to apply the trimming technique to predict the adoption in the future [56]. Generally, farmers are confident to trim because it may improve their livelihoods and due to the availability of information on trimming [57]. In our case, the information brought by the UB-UKCEH-ICRAF personnel is understood to dictate that trimming is only for the lower dry branches on the trees to a height of 10–12 m. However, in practice, farmers are uncomfortable trimming to 12 m. The data show that most farmers were seniors, which may have physical and health constraints. The old farmers were only able to trim three to five trees per day roughly, as revealed in the interview sessions; and only one to three trees for those above 7 m. Furthermore, farmers were unsure of their ability to perform trimming over a large area, with an area of one hectare containing 600 trees, regardless of their long experience of farming (over 20 years). In summary, it is easy for farmers to understand the trimming procedures, but not to practice it. Furthermore, the non-self-efficacy factor—that is the combination of externality aspects [33,58]—is also a significant consideration for intentions. We found that trimming is perceived to be expensive (when done by somebody else) and time-consuming; though having ecological benefits.

Hence, the trimming technique is perceived as complicated because of sound technical problems and limited supports from the stakeholders, regardless of the plant productivity and ecological benefits. Hence, the question is with the farmers" resource scarcity and social supports, will they be willing to apply the technique in the future? Discussions with the farmers revealed suggestions for the option of loans (from the forest management) to pay workers to trim. The other alternative is to using machinery to fasten the trimming process, which is probably cheaper, faster and safer. The use of machinery is something that researchers will investigate as part of any future forest management strategy, for example, telescopic pruners can reach 8 m in height. In-depth discussions reveal an interesting thing; farmers had proposed alternative techniques beyond trimming—such as pine thinning by removal of selected trees. The advantages proposed by farmers were (a) increasing light intensity for coffee plants (just as the trimming); (b) no need to add fertilizers; (c) no need to climb pine trees at extended heights; and (d) cheap and easy to do. A leading farmer experienced that without thinning, the soil fertility and strength of pine trees would decrease with time, as exemplified at a certain forest location (identified as plot #84); plant growth as stunted and at the risk of falling during heavy rains. He argued that it would be dangerous for individuals under the trees and considering that the forest is an also educational and recreational destination. He recommends that thinning needs to be accomplished every 10–15 years. The research team also heard of other farmers" innovative techniques on agroforestry management that are beyond our study at this stage.

## 5. Implications

Based on our experience at UB Forest, the trimming innovation has a prospect to be adopted, but with caution. On one side there is a sound theoretical basis that trimming can increase the productivity of pine–coffee agroforestry and improve the microclimate, suggesting the innovation has productivity-ecological benefits. However, it may be hampered by limited support and scarcity of farmer resources and conditions (such as old age), indicating issues on psycho-social burdens and practicality. The theory of planned behavior is a useful theoretical lens to identify the opportunity and obstacles of applying trimming techniques from the farmer's point of view. Furthermore, we have shared these issues with stakeholders (researchers, local government officers, UB forest management, key farmers, etc.) through a series of seminars. As a reflection, the attendance and engagement of the stakeholders including the government agents were high at every seminar and showing support to the research team and the findings of this study.

For the implications, the trimming program, which has been employed over two years, we have highlighted options: keep the innovation by (1) providing subsidy or loans for the farmers to

hire workers to trim or buy trimming machinery, and/or (2) integrate with the farmer techniques. The first option has conditional acceptance because of perceived financial and technical burdens. The second option has a higher chance of acceptance and implementation by farmers, first, for one reason, it would better align with the farmer's needs and capabilities. For other reasons, it would also enhance collaboration between researchers and farmers. Both options, however, have a challenge of how to cooperate with the stakeholders. For this reason, the plan and knowledge learned from action research need to be shared with the stakeholder from the beginning of any agroforestry innovation development. To reflect, this experience has improved the understanding of researchers and the collaborative options that would enhance the capacities for all the social actors in the agroforestry system. Based on this research we would encourage the application of the second option in the future.

We learn that technical constraints (the majority were senior farmers and low income) and lack of optimal external support may be important limitations in the future if there are no specific interventions. The new intervention should be not only on technical assistance, but also the sharing of information or knowledge and collaborations between stakeholders. To sustain, the communicative intervention processes require a participatory and integrative approach. Co-management of agroforestry innovations may be developed throughout the stages, from the creation, pilot/local implementation and scaling up. We need to explore any aspirations and local knowledge, contemplating the specific demographic conditions and needs of the marginal farmers at the UB Forest and the demand for stakeholder integration. Our experience may be useful as a lesson learned for policymakers and executives to comprehend the social-ecological dynamic of agroforestry management, let alone the Javanese farmers who reside close to forest lands.

As a reflection, this action research is useful, but needs improvement. We notice that the process of transferring agroforestry innovation cannot only consider innovation and farmers, but also the forest management authority. The key to accelerating the innovation transfer process is the use of an integrated communication and extension strategy. For this reason, it is necessary to establish connections between external scientists as initiators of innovation, farmers as practitioners and relevant authorities [59,60], in the case of our study is the UB Forest management. Moreover, a study shows that farmers" knowledge is often overlooked in the transfer of innovation [61,62]. In our case, farmers can observe biophysical problems in the context of their socioeconomic complexity. A farmer revealed:

> *"Regarding the soil . . . the farmers should fertilize it every six months . . . the nutrients have been reduced because [it has been absorbed by] pine roots. I have tried like that [fertilizing], and then I calculated the maintenance costs. I calculated and the results were even. Finally, I let it go, I did not care—'simalakama'!"* (emphasis added).

Simalakama is the name of a bitter fruit for traditional medicinal ingredients. People usually say, "as if eating simalakama fruits," a local parable to express a dilemma when someone wants to do something or not.

More than this, farmers—through various considerations—have their rationality and solution preference. The results of in-depth interviews and FGDs show that outside of trimming, farmers have crop management preferences through thinning techniques. After 10 to 15 years, it is believed that pine plants need to be sparsely spaced, with a spacing of twice the current one (currently in $2 \times 3$ m). Farmers recommend a $2 \times 6$ m spacing for pine trees because with more incoming sunlight, then—according to farmer experience—there are simultaneous benefits. The advantages for productivity and ecology are: "soil fertility is increasing"; "reduced acidity"; "better coffee plant growth"; "bigger plant"; "sturdy, when exposed to the wind the plants will be stronger" and "groundwater storage is better". For social benefits, farmers framed it as: "both [parties] can get [benefits]; forestry gets, the community gets too." Other techniques, vice versa, jargonized as "*Sana untung, sini buntung*" [the party over there gets profits, the community is in disadvantaged]" (results of in-depth interviews and an FGD with farmers).

Consequently, the transfer of innovation does not only come from scientists focused on productivity-ecological aspects [63], but also those embracing the aspirations of farmers and the interest

of powerholders. Therefore, knowledge of practical and political interests is needed simultaneously. This means that various modes of innovation transfer can be combined: from scientists (linear, top-down), farmers" needs (bottom-up) and interests from politics/UB Forest (lateral). Since its inception, bidirectional extension and communication strategies [64] or participatory strategies need to be developed although previous studies have shown that it is not easy for change agents to do so [6]. Included in the method action research is the need to apply transdisciplinary research, where farmers" local knowledge needs to be examined, confirmed and incorporated with those from scientific knowledge [65]—as an agroforestry knowledge pool. To sum up, efforts to balance the distinct roles of the tripartite agents for profit—prove-it—prophets may be more easily achieved through the participatory-collaborative approach [56,66].

Specific types of land management contracts between forest authorities and farmers looking for land to grow their food crops became known in 19th Century southeast Asian teak growing traditions under the name "taungya"—originally referring to indigenous swidden-fallow practices. The current pine–coffee management contracts represent considerable development since the earlier schemes [14,67], but the unequal power distribution of the past is still loading the dice against schemes that are both fair and efficient.

## 6. Conclusions

Based on the study, the adoption of the trimming innovation is prospective, though it needs adaptations. On one hand, on some levels trimming is believed to increase the productivity of pine–coffee agroforestry and maintain the microclimate, suggesting the productivity-ecological benefits of the innovation. However, this may be hampered by limited support and scarcity of farmer resources and conditions, indicating issues on psychosocial burdens and practicality. We have shared these issues with stakeholders (researchers, local government officers, UB forest management, key farmers, etc.) through a series of seminars. Practically, the trimming program that has been managed for two years, has two main options: keep the innovation by providing subsidy or loans for the farmers, and/or integrate with farmer techniques. Further options for innovation are necessary, including direct pruning of coffee and crop fertilization strategies, some of which are being tested now.

We learned that technical constraints (most senior farmers and low income) and lack of optimal external support may be an important limitation in the future if there are no specific interventions. a new intervention should be not only on technical assistance, but also the sharing of information or knowledge and collaborations between stakeholders. Sustaining the communicative intervention processes requires a participatory and integrative approach. Co-management of agroforestry innovations may be developed throughout the stages, from the creation, local implementation and scaling-up. This approach is crucial because stakeholders have roles and specific rights or interest in making decisions. Moreover, we had limited initiatives from farmers to be integrated with agroforestry innovation development. For these reasons, we need to explore more any local knowledge and practices to meet with the specific conditions and needs of the marginal farmers and their unique relations with forest management. Considering that this study had only a relatively small number of participants, a larger number of participants is needed to increase the generalizability of the results. However, our experience may be useful as a lesson learned for policymakers and executives to comprehend the social-ecological dynamic of agroforestry management, as well as the Javanese farmers who reside adjacent to forest lands.

**Author Contributions:** E.D.C.; M.v.N.; N.M. and R.R. conceptualized the study. E.D.C. designed the study, while S.F. and D.W. collected quantitative data; E.D.C. gathered qualitative data as well as acted as their undergraduate supervisor. Data curation and graph was prepared by E.P., D.W. and S.F. Then, E.D.C.; and M.v.N. assisted by E.P.; S.F.; and D.W. wrote the original manuscript preparation. M.v.N.; N.P.M.; R.L.R. and E.D.C. reviewed, added and edited the final manuscript. All authors have read and agreed to the published version of the manuscript.

**Funding:** This research received funding from Faculty of Agriculture Brawijaya University, Grant number DIPA-042.01.2.400919/2019 and the the UK Natural Environment Research Council grant Sustainable Use of

Natural Resources to Improve Human Health and Support Economic Development (SUNRISE, grant number NE/R000131/1).

**Acknowledgments:** Acknowledgment is given to the agroforestry group members from the Sumbersari community at UB Forest for their generosity in participating in this study and to the stakeholders of various institutions in their involvement during research result seminars.

**Conflicts of Interest:** The authors declare no conflicting interests. The funders had no role in the design of the study; in the collection, analyses or interpretation of data; in the writing of the manuscript or in the decision to publish the results.

## Appendix A. Survey Instruments Used for Measuring Farmer Intention

| NO. | Could you please respond (V) to the following statements with one of the answer options? | Answer options<br><br>1 = strongly disagree<br>2 = disagree<br>3 = do not know (neutral)<br>4 = agree<br>5 = strongly agree | | | | |
|---|---|---|---|---|---|---|
| | | 1 | 2 | 3 | 4 | 5 |
| | I believe that pine trimming . . . | | | | | |
| 1. | . . . provides benefits. | | | | | |
| 2. | . . . increases the incoming light. | | | | | |
| 3. | . . . fertilizes the soil. | | | | | |
| 4. | . . . is necessary because the leaf litters can be made into mulch. | | | | | |
| 5. | . . . is useful because the branches can be used as firewood. | | | | | |
| 6. | . . . increases coffee production. | | | | | |
| | In my opinion . . . | | | | | |
| 7. | . . . I understand the discussion about trimming. | | | | | |
| 8. | . . . experts help farmers know how to trim through demonstration. | | | | | |
| 9. | . . . my family supports trimming. | | | | | |
| 10. | . . . discussions with friends encourage me to look for additional information about trimming. | | | | | |
| 11. | . . . the local community supports trimming. | | | | | |
| 12. | . . . hamlet personnel support trimming. | | | | | |
| 13. | UB Forest takes care farmers to take care of the agroforestry lands. | | | | | |
| 14. | UB Forest supports trimming. | | | | | |
| | I am confident to trim, because . . . | | | | | |
| 15. | . . . it is easy to understand. | | | | | |
| 16. | . . . it is easy to do. | | | | | |
| 17. | . . . it is easy to trim at 10 m high. | | | | | |
| 18. | . . . it is cheap. | | | | | |
| 19. | . . . | | | | | |
| 20. | . . . | | | | | |
| 21. | . . . it helps improve my life. | | | | | |
| 22. | . . . the demonstration by experts encourage me to trim. | | | | | |
| 23. | I will do trimming in the farming land. | | | | | |

## Appendix B. Guideline Questions for In-Depth Interviews

1.　What types of plants or crops do you grow under the pine shade?
2.　What are the responses from the forest management when you grow coffee under this shade?
3.　Give your thoughts on the pine trimming technique recommended by the experts.
4.　Please describe your experience when trimming pine.
5.　How do you feel when trimming?
6.　What obstacles do you face when you want to do trimming?
7.　Do you have any experiences other than trimming to increase your coffee production?
8.　What are the advantages of these alternative methods?

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
