# Peer review of "Agroforestry Innovation through Planned Farmer Behavior: Trimming in Pine–Coffee Systems"

_land, doi:10.3390/land9100363_

Round 1

Reviewer 1 Report

This paper presents a study of practical importance in an area of minor importance in agroforestry worldwide. However, the specific conditions and constraints are well explained (in too much detail!).

Overall the paper is too long and 'wordy'. It would be greatly improved by being cut to 33-50% with greater use of citations to provide explanation and details. Currently it reads like a student thesis and not a research paper.

The description of the agroforestry practices sound very similar to what is well known as 'Taungya' farming. However, the word is not used in the MS. The differences and/or similarities of your system to Taungya should be explained. In many parts of the tropics the Taungya system has been abandoned, so your results should be discussed in relation to this situation. This will involve substantial revision of the manuscript.

There are many places in the manuscript where word spacing is incorrect.

Specific edits:

Line 39. Insert a comma after context

Line 92. deepen = a deeper

Line 108. Delete the second 'levels'

Line 109. promoting = promotion of

Line 142. in = to

Line 516. Insert: 'as' complicated   

Author Response

Response:

Thank you for the interest in the manuscript, even though ‘it relates to an area of minor importance in agroforestry worldwide’.  We agree with that assessment if the focus would be purely on the pine-coffee association, that has limited application globally. However, we hope to convey that our approach to balancing ecological and social dimensions of innovation has wider applicability, while progress made in the principal-agent management contracts commonly called ‘social forestry’ in the past still have a way to go.

Your comment that ‘the specific conditions and constraints are well explained (in too much detail!)’ led to a recommendation to cut to 33-50% with greater use of citations to provide explanation and details. Given your own next comment on ‘taungya’, we doubt, however, that further use of citations will provide the reader with the relevant contextual understanding. We have made some cuts, but also added some further detail requested by the other reviewers and ended with a manuscript of approximately the original length.

We have now referenced the term ‘taungya’ in both introduction and discussion. We did not, however, delve deep into the history and current differences and/or similarities of the current pine-coffee systems with Taungya as historically used in many parts of the tropics the Taungya system… We instead rely on citations to earlier discussions on why these systems were abandoned (e.g. understood in terms of principal-agent theory). Instead of ‘substantial revision of the manuscript’ that you proposed to discuss current findings in relation to this situation, we kept it light, but it can be a topic for further detail in a subsequent account of the ongoing evolution of these types of management contracts.

There were indeed many places in the manuscript where word spacing was incorrect and we have addressed that. Thanks also for the specific edits suggested, we have applied them all.

Reviewer 2 Report

Your paper could be shortened. Many long and complicated sentences.

Author Response

Thank you for all the specific edits proposed, which we have accepted.  We have indeed found long and com

Daer Reviewer 2,

Thank you for all the specific edits proposed, which we have accepted.  We have indeed found long and complicated sentences that could be split-up and simplified.

Reviewer 3 Report

To revise the manuscript, authors should consider the following issues-

  1. The introduction can be shortened and I missed the coherence in the texts. Research questions are scattered, needs to be organized in one paragraph preferably at the end of the introduction. Please consider re-writing the research questions or aim of the study (Lines 79-83 and 130-131 in one paragraph)
  2. Please include a map of the study areas in your method section, preferably after line 215. What type of questions were asked to the farmers? please add a supplementary file for that. Method sections can be organized more nicely.I suggest the theoretical framework can be plotted after the introduction.In methods first talk about the study areas, data collection, data analysis and so on....
  3. In your discussion, you should discuss your findings with the help of your theoretical framework. Please add a paragraph about the limitation of the present study.
  4. Conclusion-You have mentioned cp-management as a solution. However, you did not write about the benefits of co-management and how it will help in your case study.
  5. English edit and proofreading is a must.

Author Response

Dear Reviewer 3,

We thank the third reviewer for the suggestions that helped us improve the manuscript. Specifically, we made efforts to increase coherence in the introduction.

We now end the introduction with a single paragraph on research questions and aim.

We did include a map of the study area.

We added further detail on the type of questions asked to the farmers.

We reorganized the Methods section as suggested

We did add further reference to the theoretical framework in our discussion of key findings, trying to avoid redundancy…

We did add a few sentences on the limitation of the present study.

In our conclusion on co-management we are aware of the many constraints of the historical ‘principal-agent’ context that can only be addressed in small steps. Our main message is that innovation towards further co-management in the case study area needs to be aware of the ‘path dependency’ of the current situation – a conclusion that we expect, mutatis mutandis, to be applicable elsewhere.

We made further efforts to edit the English and proofread the manuscript.
